# Neuromuscular Fitness Is Associated with Success in Sport for Elite Female, but Not Male Tennis Players

**DOI:** 10.3390/ijerph18126512

**Published:** 2021-06-17

**Authors:** Karoly Dobos, Dario Novak, Petar Barbaros

**Affiliations:** 1Pál Péter Domokos Elementary School, 1119 Budapest, Hungary; karoly.dobos93@gmail.com; 2Hungary and Department of Combat Sports, University of Physical Education, 1123 Budapest, Hungary; 3Faculty of Kinesiology, University of Zagreb, 10000 Zagreb, Croatia; petar.barbaros@kif.unizg.hr; 4Institute for Anthropological Research, 10000 Zagreb, Croatia

**Keywords:** agility, explosive power, first quick quickness, flexibility, neuromuscular fitness

## Abstract

Background: The purpose of the study was to examine whether neuromuscular fitness contributes significantly to the success of eAlite junior tennis players of differing ages and sexes. Methods: The 160 participants, who were elite Hungarian junior tennis players (aged 11–17), were separated into four groups within this study, and 10 different types of field tests were used. Results: A moderate significant correlation was found between the results of the 5 m run (r = −0.42; r = −0.45), standing long jump (r = 0.39; r = 0.56), overhand ball throw (r = 0.44; r = 0.53), serve (r = 0.39; r = 0.64), amount of push-ups in 30 seconds (r = 0.32; r = 0.48), 10 × 5 m run in a shuttle run (r = −0.34; r = −0.45), the spider run (r = −0.34; r = −0.52), and competitive tennis success among U14 and U18 girls. A significant correlation between the overhead medicine ball throw test value (r = 0.47) and the current competitive performance was found only among U18 elite female tennis players. In contrast, no correlation was found between the values of the U14 and U18 male tennis players and their current competitive performance. Conclusions: Additional studies are needed to identify interventions that can increase sport-specific neuromuscular fitness with the ultimate goal of achieving better performance.

## 1. Introduction

Tennis has undergone dramatic changes in the past few decades. The technique, spin of the ball, speed of the game, anthropometric characteristics of the players, playing style, and strategy have developed significantly thanks to modern equipment, the modern court surface, and the precise selection and modern training methods. As a result, the physical fitness of tennis players has greatly increased in not only senior players, but junior players as well [1,2].

Tennis requires open skills and is an intermittent sport characterized by its repeated high-intensity efforts (i.e., first quick step, accelerations, decelerations, changes of direction and strokes, etc.) during a variable period of time (i.e., on average 90 min) [1,2,3,4]. Speed of the ball and movement of the players has an effect on both the upper and lower parts of the body. This is due to the players being required to execute movements and strokes of different directions and distances within a very short period of time. Furthermore, they are also required to adjust the movement and parameters of the ball trajectory. The average sprint distance performed in tennis is 5–7 m during an individual point, with an average of four to six changes of direction [2]. Movements in tennis can be characterized most frequently by the explosive power and the combination of medium to high aerobic and anaerobic capacity (endurance) [3,4]. This explosive power (that involves a stretching–shortening cycle) manifests itself during the rally through the number of repeated shots and tennis-specific footwork which occurs [5]. The average number of shots during a rally can be between five and six, with any given match lasting between 1–5 h depending on the type of competition [2,4]. Moreover, tennis requires the ability of a high level of coordination and proper dynamic flexibility due to the need to generate ball speed in extreme body positions [5]. Based on these facts, tennis players have to possess a high level of speed, agility, flexibility, and explosive power combined with well-developed aerobic and anaerobic endurance. However, the role of these physical variables in current competitive performance has to be examined especially in elite junior tennis players of differing ages and sexes, as they are at the beginning of their careers, and their physical characteristics, chronological age, and maturity differ from those of the adult professional players. It is well documented that the largest differences between juniors and professionals were in physical demands of play [3]. Professional players play longer matches, produce more shots, and cover more distance per match than junior players [3]. As far as measuring the physical variables, the current sport literature differs in both field and laboratory tests. Laboratory tests have high validity and authenticity and are very expensive, this being due to the quality and accuracy of the results that they can provide. However, field tests also model special movement patterns accurately, with physical variables and metabolic processes (compared to other tests) then being carried out in the laboratory, both of which characterize this sport. Furthermore, they are relatively simple and cost-effective and can be used for a high number of subjects [3].

In order to provide support in the development of young and talented athletes, it is essential to understand the factors that can have a positive influence on their sporting success. There is some evidence on the role of physical variables in the competitive performance of junior tennis players with the help of field tests [6,7,8,9,10,11]. In line with the above-mentioned, we hypothesized that neuromuscular fitness may be associated with sport success at the junior level. However, few studies have simultaneously examined the contribution of different neuromuscular factors to sport performance.

Accordingly, in the present study, we investigated the association of neuromuscular fitness with competitive tennis success among a sample of elite junior tennis players.

## 2. Materials and Methods

### 2.1. Participants

In total, 160 Hungarian junior tennis players (80 boys and 80 girls) aged between 11–17 took part in the study, with each participant having between 3 and 7 years’ experience in international and national tennis competitions. Each participant had also played 40–60 matches per year and was ranked among the top 40 in Hungary in their respective age groups, this being the acceptance criteria for the participants to qualify to be a part of this study. The program G-power (version 3.1.9.2; Heinrich Heine University Dusseldorf, Dusseldorf, Germany) with the statistical power of 0.95 was used for estimating the appropriate number of participants. Subjects were selected through the stratified random sampling method.

The 4 age groups of the present study were formed in accordance with the United States Tennis Association guidelines [5]. Their chronological age was: mean ± SD = 12.44 + 1.35 years (boys U14, *n* = 40); mean ± SD = 12.50 + 1.37 years (girls U14, *n* = 40); mean ± SD = 16.16 + 1.57 years (boys U18, *n* = 40); and mean ± SD = 16.24 + 1.67 years (girls U18, *n* = 40).

### 2.2. Design and Procedures

Ten different types of field-tests (Table 1; for more details, please check Appendix A) were applied based on previous literary data and research [1,3,5,12,13,14,15,16].

The current competitive performance variable (elaborated and used by the Hungarian Tennis Federation for several years) was also used, which represented the average values of the points won in matches by each participant. A familiarization session was performed 1 week prior to the testing, where the participants were informed in both oral and written form about the aim, the testing process, and the testing protocol. No intense physical activity was performed by the players 48 h prior to the tests. The tests were then carried out on an indoor clay court, certified by the Hungarian Tennis Federation according to Hungarian and international rules and regulations, at a temperature of 15–25 °C. The players had to execute the tests in a given order (Table 1) after a 15-min standard warm-up, which consisted of 5 min of jogging; dynamic flexibility exercises; forward, backward, and sideways running; acceleration running; and the execution of fifteen flat serves. There was a period of 4-min between the warm-up and the tests, with there being a rest period of 2 min between the trials and a 25 s rest period between the serves; this was in order to limit the effects that fatigue may play on the results. The players had 3 trials in the 5 m run (R5), standing long jump (SLJ), overhead medicine ball throw (OMBT), overhand ball throw (OBT), 8 trials in the serve (S), 2 trials in the hexagon (H), sit and reach (STR), 10 × 5 m shuttle run (SH 10 × 5), spider run (SR), and the push-ups in 30 s (PU 30 s) in the testing process. Neither the measuring equipment nor the participants carrying out the work were modified in the repeated examinations. For the R5, SH10 × 5, and SR tests, the GUR-1 electric timer (within 0.01 seconds of accuracy) was used, and for the H and PU 30 s tests, the Casio stopwatch (within 0.01 s of accuracy) was used. During the measuring of the OMBT and OBT and SLJ tests, a 1 kg stuffed ball, a 103 g small ball (diameter 8 cm), and a calibrated tape measurer (marked at every cm) were used. For the S test, the “Stalker ATS II” serve speed measurer (within ±3 km/h of accuracy) and 53–56 g and 6.5 diameter “Slazenger Ultra Vis” balls as well as at the STR test 32 cm high, 45 cm wide, and 55 cm long box were used. The study met the guidelines of recommendations of the Declaration of Helsinki [17] and was approved by the Ethical Committee of Public Health Division of the Budapest Government Office (permission # 7878/2014). Furthermore, all the players had a medical screening and received declarations of consent from their parents prior to the study.

### 2.3. Statistical Analysis

The normality of the data was checked by the Shapiro–Wilk W test. In addition, the ICC was calculated to determine the reliability of the measurements (except S test) using the Spearman–Brown formula of calculation. The correlations between the physical variables and the current competitive performance were calculated with the Spearman’s rank correlation methods. The correlations were considered as weak (|r| < 0.3), moderate (0.3 < |r| < 0.7), or strong (|r| > 0.7) [18] at a significance of *p* < 0.05. The statistical analysis of the data was carried out with the SPSS 21.0 software.

## 3. Results

Most of the data (H, R5, SLJ, SH10 × 5, SR, and PU 30s) did not fulfill the requirements of normal distribution, which was checked by the Shapiro–Wilk W test (*p* < 0,05); thus, at every variable, the basic statistical data were described as mean ± standard deviation and minimal and maximal result (Table 2). For all tests, an ICC of 80–94 was found and therefore all tests were considered to be reliable (Table 3). A moderate significant correlation was found between the R5, SLJ, OMBT, OBT, S, PU 30s, SH 10 × 5, and SR test values and the current competitive performance in girls, but not in boys (Table 4). A moderate significant correlation was found between the R5, SLJ, OBT, S, PU 30s, SH 10 × 5, and SR test values and the current competitive performance in the U14 and U18 female junior tennis players. Furthermore, in the U18 elite female junior tennis players, the OMBT test’s value showed a moderate significance of correlation with the current competitive performance (Table 5.). The H, OMBT, and STR test values in the U14 females, H and STR test values in the U18 females, and all the test values of the U14 and U18 elite junior male tennis players did not show any correlation with the current competitive performance (Table 5).

## 4. Discussion

This study aimed to investigate the association of neuromuscular fitness in sporting success among a sample of elite junior tennis players. With regard to the study aim, we can conclude neuromuscular fitness is associated with sporting success in females but not male elite junior tennis players.

There is a positive role of physical fitness on the competitive performance of junior tennis players [6,7,8,9,10,11]. Comparing the present results with previous literature is difficult, as previous studies were conducted with a different sample size, participants of a differing sex, and different test protocols, although the results of the present study (a moderate correlation between the physical variables and the current competitive performance) of Hungarian junior female players reinforce the results of the previous research [6,7,8,9,10,11]; except for the H and the STR tests, this was not the case for the male Hungarian junior tennis players. Tennis is a tactical and technically dominant sport that can be characterized by the complex interaction of physical abilities and metabolism processes from a conditional point of view [3,6,7,10]. It is well documented that the sexual maturity of female adolescents is happening earlier compared to males of the same age, with females reaching their peak physical performance much earlier [19]. We can hypothesize that increases in mass and stature have a large influence on their physical fitness measures, whereas in junior males there are some qualitative differences in performance due to other factors (i.e., technical executions, tactical solutions, diversity of shorts). Elite junior male tennis players who possess better physical fitness than their peers cannot necessarily translate this advantage into a significantly higher chance of winning. In addition, it is possible that the examined junior male tennis players can play in a more versatile way than females. This meaning that during the game, due to slightly different playing style, the junior female tennis players prefer those simpler technical and tactical solutions in which their physical abilities (i.e., explosive power) can play a more significant and effective role.

However, it is surprising that the better H test value of the examined junior tennis players did not result in a better current competitive performance, which contrasts with previous studies [20]. The results observed may be due to several factors including a uniformly high level of fine motor co-ordination abilities, dynamic postural control, and balance of the examined players. Additionally, the movement patterns with the test are not related to the typical movements in tennis. The value of the STR test (which is often an official part of any assessments protocols) did not show a correlation with current competitive performance either in the examined junior tennis players, which might be explained by the fact that static flexibility performance could not be transferred to dynamic situations in tennis [21]. Nonetheless, static stretching was considered the safest and best method to improve a players’ flexibility and decrease the risk of injury.

### 4.1. The First Step Quickness, Acceleration, and Agility

The R5 test value showed a moderate correlation with the current competitive performance in U14 and U18 females. This test measures the speed of the first step in running/sprinting forward. In tennis, the emphasis is on the speed of the first step in order for the player to reach an effective hitting position and how fast the player can get into motion, overcoming the inertia of his/her own body. The 5 m straight run models the launching of this motion effectively for observation [8]. Although the speed and frequency of the strokes executed by the junior female tennis players lag behind those of adult female players, especially in the U14 age group, the first step and the ability to speed up within a short distance are both essential requirements for handling the ball correctly to successfully solve the game situations [2]. Tennis is defined as a sport with continuous changing situations, as every ball hit or received has a different speed, position, and spin. It is important to point out that the functional mechanism for speed and agility is different. The way in which a player moves around the court determines the successfulness of the player [7,8], this being due to the fact that proper footwork makes for effective and successful execution of the strokes. The results show that the SH 10 × 5 and the SR test values in the U14 and U18 elite junior female tennis players show a moderate correlation with current competitive performance. The nature of tennis having fast exchanges requires a high level of linear and multi-directional agility, which is why explosive and controlled moves, direction changes, fast instances of slowing down, and proper support are also important requirements of competing successfully at this age, in addition to forming the basis of the technical execution of accurate and successful strokes.

### 4.2. Lower Body Explosive Power

The obtained data show that the explosive power of the lower limbs (SLJ test) in the U14 and U18 elite junior female categories show a moderate correlation with current competition performance. The contact of the legs with the ground creates a so-called starting force that provides the basis of all tennis strokes. Several studies have explored the correlation between the explosive power of the lower limbs as well as the sprinting of various distances [9], and the correlation between the lower body explosive power and running speed, modified by direction changes [22,23,24]. This is why the development of the lower body explosive power is of utmost importance in the preparation of elite junior players. A tennis player who is able to exert explosive power is also able to move quickly around the court and execute strokes with effective speed [22,23].

### 4.3. Upper Body Explosive Power

A moderate correlation was found between the values of OBT test of the U14 elite junior female players and the U18 elite junior tennis players and the OMBT test value of the U18 players and current competitive performance. The stretch and shortening cycle (the plyometric movements) is the most frequent contraction type in tennis, as the coordination pattern of most of the strokes is built up from this specific contraction. This is why the use of plyometric throwing exercises is indispensable in developing the most beneficial stroke power [1,3,13], with special consideration given to those athletic throwing forms (overhand ball throw, medicine ball throws from overhead and forehand and backhand side) that aid in the learning and perfecting process of the technical elements that occur in tennis.

### 4.4. Tennis-Specific Explosive Power

In professional tennis, the role of the serve being key is unquestionable. The serve is the only technical element that is executed by the player independent of the opponent’s ball. This independent execution ensures the highest possible level of movement control by an individual player. However, the serve does not play such a dominant role in the game of junior female players, especially in the U14 age group. Nevertheless, the S test value of U14 and U18 elite junior female tennis players showed a moderate correlation with current competition performance, with the reason for that being the faster the serve hit, the shorter preparation time available for the receiving player. This is why the chance of the server to gain points increases, but that of the receiver decreases with high-speed serves, as the receiver has to match their movements to the trajectory parameters of the ball within a very short time, which is a great challenge in junior-level tennis due to the players having not yet perfected their skill set. At the same time, it must not be forgotten that the speed of stroke is only one factor determining the quality of the serve (reliability, assurance, accuracy, spinning, and speed). It is important to point out that the technical execution of the serve is an important factor affecting speed, and high-level competition performance cannot be completed without a reliable and accurate serve in the modern game [25].

### 4.5. Muscle Endurance

The obtained data show that the muscular endurance (PU 30s test) in U14 and U18 elite junior female players shows a moderate correlation with current competitive performance. The pectoralis and the muscles in the arm executing the stroke show great activity in the phases of speeding up the racket while executing different technical elements [26]. As well as this, there is also a high, repeated, and unbalanced load on the upper body of junior tennis players. The matches can last many hours, and during this time, a player may execute several hundred strokes; accordingly, if they want to maintain the level of their stroke quality whilst avoiding the risk of injury, they should develop the explosive power and muscle endurance required to do so.

This study has a number of limitations, which will be discussed below. Firstly, the subjects involved in this study were highly selected youth tennis players in a very sensitive and crucial developmental phase. Secondly, we did not evaluate the biological age of the participants, which is known to influence neuromuscular performance; thirdly, we did not have the possibility to look at the anthropometric variables and well as at the tennis-specific variables like technical execution of the serve or forehand/backhand medicine ball throw; and fourthly, we did not have the possibility to look at the mental and physical fatigue that may have occurred during the testing process; therefore, it could have potentially affected the most effective movement execution.

In summary, the present study suggests that neuromuscular fitness is associated with competitive tennis success in female junior tennis players, but not male elite junior tennis players. Our findings provide useful information for coaches to create a wide range of tennis-specific situations to develop a proper performance, especially for their player’s neuromuscular fitness. Additional studies are needed to identify interventions that can increase sport-specific neuromuscular fitness with the ultimate goal of achieving better performance.

## 5. Conclusions

The result of the present study showed that the better performance in selected field tests (except for H, OMBT, and STR in the U14 girls, H and STR in U18 girls) showed a better current competitive performance in the U14 and U18 elite Hungarian female tennis players, but in contrast, U14 and U18 elite Hungarian male players did not show a better current competitive performance in selected fields. We can hypothesize that sexual maturation has a large influence on female physical fitness measures, and due to slightly different playing style, the importance of neuromuscular fitness is greater. In males, there are some qualitative differences in performance due to other factors. In addition, future research should focus on complex measuring of different performance variables that happen during a match or training (i.e., by taking the speed and accuracy of hitting on the serve, forehand, and backhand strokes as a criterion or indicator of competitive level). Furthermore, more research is required in order to develop a tennis-specific field test that can measure the key physical variables more precisely. Finally, future research should be extended to other age categories, taking into account biological age rather than chronological age.

## Figures and Tables

**Table 1 ijerph-18-06512-t001:** The sequence and variables of selected field test.

Code	Tests	Variables Measured
**H**	Hexagon (0.01 s)	Agility and coordination
**R5**	5 meter run (0.01 s)	Acceleration and speed of first-step
**SLJ**	Standing long jump (m)	Explosive power
**OMBT**	Overhead medicine ball throw (m)	Explosive power
**OBT**	Overhand ball throw (m)	Explosive power of the dominant side of the body
**S**	Serve (km/h)	Tennis-specific explosive power
**PU30s**	Push ups in 30 s (freq.)	Muscle endurance
**SH10 × 5**	10 × 5-m shuttle run (0.01 s)	Linear agility
**SR**	Spider run (0.01 s)	Multidirectional tennis-specific agility
**STR**	Sit and reach (cm)	Flexibility

**Table 2 ijerph-18-06512-t002:** The descriptive statistics of the elite junior male and female tennis players (*n* = 160).

Tests	U14 Girls *n* = 40		U14 Boys *n* = 40		U18 Girls *n* = 40		U18 Boys *n* = 40	
	Mean ± SD	Min–Max	Mean ± SD	Min–Max	Mean ± SD	Min–Max	Mean ± SD	Min–Max
**H (0.01 s) ***	12.01 ± 1.12	1.34–14.52	12.18 ± 2.00	10.00–20.00	10.90 ± 0.84	9.14–2.40	10.76 ± 1.18	8.84–13.53
**R5 (0.01 s) ***	1.29 ± 0.07	1.15–1.44	1.27 ± 0.06	1.15–1.41	1.25 ± 0.04	1.16–1.38	1.14 ± 0.05	0.99–1.27
**SLJ (m) ***	1.69 ± 0.19	1.30–2.08	1.82 ± 0.16	1.30–2.21	1.85 ± 0.16	1.42–2.20	2.23 ± 0.23	1.72–2.88
**OMBT (m)**	8.84 ± 1.90	5.47–13.62	9.35–2.14	6.09–14.32	10.74 ± 1.61	7.88–14.48	14.50 ± 2.41	9.20–19.20
**OBT (m)**	27.03 ± 5.69	16.57–39.90	35.42 ± 6.20	25.75–50.30	30.65 ± 5.97	19.80–44.30	48.23 ± 7.54	32.36–66.8
**S (km/h)**	128.62 ± 18.42	87.00–176.00	140.52 ± 17.07	110.00–175.00	152.00 ± 10.31	133.00–169.00	174.60 ± 13.8	144.00–211
**PU30s (freq.) ***	8.66 ± 5.90	1–25.00	17.42 ± 4.51	9.00–28.00	10.89 ± 7.99	1.00–30.50	27.23 ± 8.05	11.00–44.00
**SH10 × 5 (0.01 s) ***	20.65 ± 1.47	18.48–26.40	19.95 ± 0.95	18.00–21.70	19.51 ± 0.83	18.04–21.40	18.52 ± 078	17.08–20.20
**SR (0.01 s) ***	21.80 ± 1.64	18.56–26.13	20.81–1.08	18.76–23.79	19.28 ± 0.94	17.10–21.90	18.05 ± 0.97	15.96–20.00
**STR (cm)**	20.18 ± 7.42	1.32–34.00	13.57–6.45	1.00–31.00	23.72 ± 7.04	9.00–37.00	17.56 ± 8.67	1.00–35.00

Legends: H—hexagon; R5—5 m run; SLJ—standing long jump; OMBT—overhead medicine; ball throw; OBT—overhead ball throw; S—serve; s; PU30—push ups in 30 s; SH10 × 5—10 × 5 m shuttle run; SR—spider run; STR—sit and reach; Shapiro–Wilk W test (* *p* < 0,05).

**Table 3 ijerph-18-06512-t003:** Reliability values of selected field test for elite junior tennis players.

Tests	H	R5	SLJ	OMBT	OBT	S	PU 30 s	SH 1 × 5	SR	STR
ICC (95% CI)	0.91 (0.90–0.92)	0.93 (0.92–0.94)	0.82 (0.80–0.84)	0.89 (0.97–0.91)	0.86 (0.85–0.87)	0.87 (0.85–0.87)	0.85 (0.80–0.90)	0.87 (0.85–0.90)	0.90 (0.89–0.91)	0.92 (0.90–0.94)
CV%	3.1	2.3	3.5	3.9	3.8	2.3	3.8	2.9	2.1	1.5

Legends: H—hexagon; R5—5 m run; SLJ—standing long jump; OMBT—overhead medicine ball throw; OBT—overhead ball throw; S—serve; PU30s—push ups in 30 s; SH10 × −5—10 × 5 m shuttle run; SR—spider run; STR—sit and reach; ICC—intraclass correlation coefficient; CI—confidence interval; CV—coefficient of variation.

**Table 4 ijerph-18-06512-t004:** Spearman correlation coefficients between physical variables of selected field tests and the current competitive performance of the same sex as a single group (*n* = 160).

Groups	H	R5	SLJ	OMBT	OBT	S	PU 30 s	SH 10 × 5	SR	STR
Girls	−0.16	−0.43 *	0.50 *	034 *	0.49 *	0.46 *	0.39 *	−0.41 *	−0.39 *	0.04
Boys	−0.19	−0.20	0.11	0.11	0.17	0.21	0.06	−0.21	−0.16	0.11

Legends: H—hexagon; R5—5 m run; SLJ—standing long jump; OMBT—overhead medicine ball throw; OBT—overhead ball throw; S—serve; PU30s—push ups in 30 s; SH10 × −5—10 × 5 m shuttle run; SR—spider run; STR—sit and reach (* denotes significant correlations at * *p* < 0.05).

**Table 5 ijerph-18-06512-t005:** Spearman correlation coefficients between physical variable of selected field tests and the current competitive performance in elite junior tennis players (*n* = 160).

Groups	H	R5	SLJ	OMBT	OBT	S	PU 30 s	SH 10 × 5	SR	STR
U14 girls	−0.06	−0.42 *	0.39 *	0.29	0.44 *	0.39 *	0.32 *	−0.34 *	−0.34 *	0.08
U18 girls	−0.11	−0.45 *	0.56 *	0.47*	0.53 *	0.64 *	0.48 *	−0.45 *	−0.52 *	0.01
U14 boys	−0.15	−0.08	0.10	0.03	0.17	0.03	0.05	−0.22	−0.26	0.35
U18 boys	−0.11	−0.18	0.11	0.05	0.01	0.25	0.18	−0.16	−0.05	0.02

Legends: H—hexagon; R5—5 m run; SLJ—standing long jump; OMBT—overhead medicine ball throw; OBT—overhead ball throw; S—serve; PU30s—push ups in 30 s; SH10 × −5—10 × 5 m shuttle run; SR—spider run; STR—sit and reach (* denotes significant correlations at * *p* < 0.05).

## Data Availability

The data presented in this study are available on request from the corresponding author.

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
