# Peer review of "Neuromuscular Fitness Is Associated with Success in Sport for Elite Female, but Not Male Tennis Players"

_ijerph, 2021, doi:10.3390/ijerph18126512_

Round 1
Reviewer 1 Report
Neuromuscular fitness is associated with success in sport for elite female, but not male tennis players
General comments
The present paper deals with neuromuscular fitness and its relationship with the success of Elite Junior Tennis Players of Differing Age (14 & for boys and girls, 18 & for boys and girls) and genders. The study was conducted on 160 juniors who completed 10 tennis specific tests on the court. Significant correlations were found between the scores of the specific tennis tests. The correlation coefficients indicated that physical variables are only one segment of the complexity of Current Competitive Performance.
Specific comments
Introduction
Lines 28-65: As these are young tennis players, I suggest that the authors define more precisely (within the available data) the characteristics of young players' tennis (duration of the match, sets, games, point, change of direction of movement, number of strokes, tempo ...) and not only the values that apply to older elite players.
Lines 29-34: Support your claims about changes in tennis play with appropriate references.
Line 57: I suggest replacing the term "samples" with "subjects".
Line 64: Sporting Success is not an appropriate term. I suggest "competitive tennis success" or "tennis success".
Materials and methods
Line 82: And Table 1, I suggest changing "horizontal force" to "explosive force" for the standing long jump test. I also suggest changing the name "measurand" as it is a technical item. The velocity of the power is largely dependent on the application of biomechanical principles and the quality of the movement pattern. The authors should also back this up with references.
Line 107: 2.3 Since most of the tests are common knowledge and the authors have made only minor adjustments, the "Description of selected field tests" can be classified as an appendix.
Line 175: The authors should explain why they did not do the regression analysis and check to what extent individual tests explain the variance of the criterion variable.
Results
Line 204: The legend "Overhead Ball Throw" is contained in two lines. I assume that this is a typo.
Line 210. the text should be above the table - reach.
Discussion
Lines 216-230: I suggest that the authors reconsider the explanation in this section. First, with regard to only one of the motor skill domains (Explosive Power), it is possible that this domain does not have a significant impact on competitive performance despite some correlations with competitive performance. Second, due to the aforementioned complexity of tennis, the importance of motor skills decreases with the age of the players. Thirdly, in female players, due to a slightly different playing style (otherwise this has not been demonstrated in scientific articles), the importance of explosive power is greater.
Line 231: The conclusion that Hexagon is not suitable for competition because the content itself is not related to the typical movements in tennis.
Line 234: At this point, the authors could explain why the flexibility test was included in the battery that measures explosive power.
Line 239: The authors should emphasize that the R5 test measures the speed of the first steps in running/sprinting forward. In tennis, most movements are performed later, so the type of movement is different than in the test. It is also important that the authors point out that the functional mechanism for speed and agility is different.
Line 261: The authors should support the claim with references.
Line 269: The authors mention simulating forehand and backhand strokes with the drug. Why was this test not included in the study?
Line 283: Depending on the complexity of the serve, the technical execution of the serve is an important factor affecting speed. The authors should support this with references and an explanation that they did not measure this in the study.
In this chapter, all studies on constraint need to be stated.
Conclusions
Lines 305-312: I suggest that this chapter be rewritten depending on the comments on chapter - 216-230.
Author Response
Dear Reviewer:
We are pleased to resubmit for publication the revised version of Manuscript entitled " NEUROMUSCULAR FITNESS IS ASSOCIATED WITH SUCCESS IN SPORT FOR ELITE FEMALE, BUT NOT MALE TENNIS PLAYERS". We appreciate your constructive criticisms. We have addressed each of your concerns as outlined in attachment.
We hope that with these modifications, our paper can now be accepted for publication.
Sincerely,
Authors

Reviewer 2 Report
Neuromuscular fitness is associated with success in sport for elite female, but not male tennis players.
The purpose of the study was to examine if the performance in various physical exercises and the speed of the tennis serve contributes significantly to the success of elite junior tennis players of differing age and sexes. For that, 160 participants junior tennis players (aged 11-17), were separated into four groups, two groups for sexes. Some moderate significant correlations were found in U14 and U18 elite female tennis players, but not in males.
General comments.
We acknowledge the work and effort of the authors in carrying out this study, but we must draw attention to several issues that, in our opinion, put the study design at risk.
The first issue is that the study, in our opinion, should have been carried out considering all athletes of the same sex as a single group. In this way, the aim of the study could have been better clarified than by doing it only in groups. This would also have provided better information about something that does not seem logical, namely that in large groups of athletes, or in any group of people in general, performance data in physical activities do not fit the normal curve. We do not know what the reason may have been, but we have not come across any such case.
As a consequence of what was previously indicated in relation to the adjustment of the data to normality, we find variables measured in seconds, metres, centimetres, and kilometres per hour whose indicators of centrality are expressed with the median, and, furthermore, without any indicator of variability. Again, we consider that with quantitative, continuous, and ratio variables such as those indicated, it is necessary to use the mean and standard deviation as descriptors of the characteristics of the variables.
Another negative consequence related to the normality of the data is the use of Spearman's rank correlation, also taking into account that sport performance has been determined by "points", not by the order in which the athletes are ranked. With this type of analysis, much information is lost, as the differences between the scores in the physical tests and in the ranking of tennis players by competitive performance are not detected.
Another important issue is that the scores given to players by the procedure described in the study are considered as a "gold standard", accepting that such a score actually ranks athletes well for their current specific performance ability. This is likely to be a major risk because the score is dependent on numerous factors that are difficult to quantify and to grade in the scoring and ranking of players. It is likely that better information on the importance of physical performance on specific performance could have been obtained by taking the speed and accuracy of hitting on the serve, forehand and backhand strokes as a criterion or indicator of competitive level. We agree that the speed of stroke, or rather of the ball, is only one factor determining the quality of the serve, but it is undoubtedly the most important, as it is what makes it most difficult to return the ball.
In relation to the reliability of the data, the ICC is presented, but there is no indication of the calculation procedure or model that has been applied, which is important, as the value of this coefficient can change a lot if the differences between measures or simply the interaction are taken into account in the calculation. But most importantly, there is no information on the coefficient of variation, which is more relevant in the assessment of reliability than the ICC, as the latter can be clearly influenced by the variability between subjects. In this sense, it is indicated that "for all tests, an ICC of .80-.94 were found and therefore all tests were considered to be reliable", and reference 17 is cited. This may be an error on our part, but it seems that this quote is not about the ICC.
We are confident that all tests were conducted with proper protocol and control, but the PU 30s test should have been controlled for body weight: more body weight would tend to decrease the result in this test, but body weight tends to have a positive relationship with ball speed on any given shot.
In this sense, a probable relevant shortcoming of the study is not having considered the basic anthropometric measurements of the players, in order to help confirm the values of the correlations obtained, and, probably, to discover a greater degree of association between the variables studied.
In relation to the above, in this type of correlational studies it is "mandatory" to check the influence of third variables in the relationships that are analysed through the appropriate partial correlations. This has not been addressed in the present study. Some anthropometric variables and especially biological maturation could have a notable influence on the relationship between physical and competitive performance variables.
In general, the discussion makes several interpretations and possible explanations for the results that can sometimes be seen as too speculative, without being based on data or references. The possible explanation for why relationships are found in women but not in men does not seem to be clear. For example, to say that "sexual maturation has a large influence on female physical fitness measures, whereas in males there are some qualitative differences in performance due to other factors", does not seem to be justified either by the present study or by previous studies. What could these factors be? Moreover, sexual maturation in women has not been analysed in this study.
A summary of the findings as follows: "the correlation coefficients indicated that physical variables are only one segment of the complexity of current competitive performance" seems rather obvious. It does not seem reasonable to propose that a tennis player's competitive performance depends exclusively on physical variables. Therefore, it would not have been worthwhile to conduct a study to reach this conclusion.
Author Response

(The authors gave the same response as above.)

Reviewer 3 Report
This paper sheds additional light on the relationship between the level of performance in tennis (ranking) and certain physical qualities. This research should be extended to other age categories taking into account biological age rather than chronological age
Author Response

(The authors gave the same response as above.)

Reviewer 4 Report
Line 78-79: Age should be presented as mean±SD. SD is standard deviation.
Table 2: Range should be presented as minimum-maximum. The range described as it is now has limited information.
Author Response

(The authors gave the same response as above.)

Round 2
Reviewer 2 Report
In my review of this manuscript I stated that according to the criticisms I indicated to the authors and the editor, in my opinion, the article did not meet the necessary requirements for publication. This is because I considered that the deficiencies observed could not be remedied except by a reworking of the work from the beginning. I am therefore sorry, but I understand that I must stand by my opinion.
Author Response
09-Jun-2021
Dear Reviewer:
Thank you for your comments. We are fully aware of your constructive criticisms. We have implemented all of your comments in our revised version of Manuscript. Also, we have addressed each of your concerns as outlined in our previous response.
We hope that with these modifications, our paper can now be accepted for publication.
Sincerely,
Karoly Dobos
Dario Novak
Petar Barbaros